# An Efficient Simulation-Based Policy Improvement with Optimal Computing Budget Allocation Based on Accumulated Samples

**Xilang Huang [1] and Seon Han Choi [2,*]**

1    Department of Artificial Intelligence Convergence, Pukyong National University, Busan 48513, Korea; huangxl901@pukyong.ac.kr
2    Department of Electronic and Electrical Engineering, Ewha Womans University, Seoul 03760, Korea
*    Correspondence: gigohan01@gmail.com

**Abstract:** Markov decision processes (MDPs) are widely used to model stochastic systems to deduce optimal decision-making policies. As the transition probabilities are usually unknown in MDPs, simulation-based policy improvement (SBPI) using a base policy to derive optimal policies when the state transition probabilities are unknown is suggested. However, estimating the Q-value of each action to determine the best action in each state requires many simulations, which results in efficiency problems for SBPI. In this study, we propose a method to improve the overall efficiency of SBPI using optimal computing budget allocation (OCBA) based on accumulated samples. Previous works have mainly focused on improving SBPI efficiency for a single state and without using the previous simulation samples. In contrast, the proposed method improves the overall efficiency until an optimal policy can be found in consideration of the state traversal property of the SBPI. The proposed method accumulates simulation samples across states to estimate the unknown transition probabilities. These probabilities are then used to estimate the mean and variance of the Q-value for each action, which allows the OCBA to allocate the simulation budget efficiently to find the best action in each state. As the SBPI traverses the state, the accumulated samples allow appropriate allocation of OCBA; thus, the optimal policy can be obtained with a lower budget. The experimental results demonstrate the improved efficiency of the proposed method compared to previous works.

**Keywords:** Markov decision process; simulation-based policy improvement; optimal computing budget allocation; stochastic system optimization

## 1. Introduction

A Markov decision process (MDP) is a discrete-time stochastic control scheme that aims to solve stochastic decision-making problems. Research on stochastic decision-making problems has been widely reported in numerous fields, such as physics [1,2], finance [3], and biology [4]. Intuitively, decision-making in MDP-based complex systems involves a process of finding an optimal policy. This has been leveraged to simulate real dynamic environments of complex systems to derive optimal solutions (i.e., policies) to predict or improve system performance; see these examples of a robot motion planning system [5], a dataflow system [6], and a mobile edge computing system [7]. The MDP consists of a set of discrete states and a finite set of actions. The MDP policy involves mapping from states to actions. When an action is implemented following the policy in a given state, it is transferred to a new state according to the transition probability and receives a reward, as shown in Figure 1. The objective of the MDP is to find an optimal policy, and the optimal policy consists of the best actions that maximize the expected sum of discounted rewards (i.e., Q-value) in each state. Since the MDP mimics practical management systems, the action space is typically large and transition probabilities are usually not known in advance; thus, directly finding the optimal policy is impractical and time consuming.

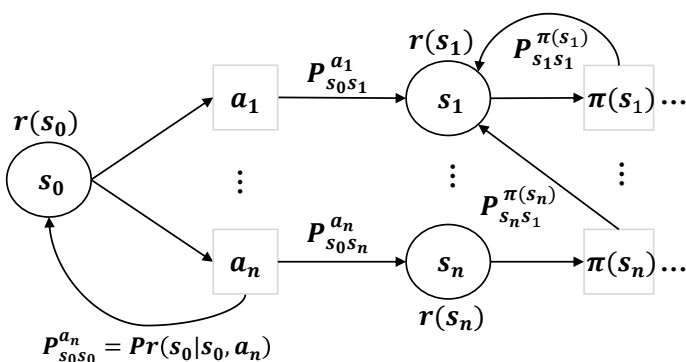

**Figure 1.** An example of MDP, where a circle represents a state and a square represents an action that is available in the state. $P$ represents an unknown state transition probability, a reward is denoted by $r$, and $\pi$ is the policy.

When it is not feasible to directly determine an optimal policy, a more appropriate solution is to improve from a given base policy, which is effective and often available in engineering practice. Simulation-based policy improvement (SBPI) (also known as rollout) [8] is a heuristic method of improving the base policy gradually by simulations. In a given state, SBPI estimates the Q-value of each action using simulations and updates the policy in the state with the selected best action depending on these values. Due to the ease of implementation of SBPI, it has been widely applied for many problems, including electric vehicle charging [9] and post-hazard recovery [10]. However, when the number of available actions in each state is large or the reachable states are numerous, each collected simulation sample may result in large variance. Thus, for selecting the best action accurately, a large simulation budget (i.e., a large number of simulation replications) is required to estimate the Q-value of each action, which results in an efficiency problem.

Ranking and selection (R&S) procedures can be used to address the above problem because they can efficiently select the best action under a limited simulation budget by allocating the budget based on statistical inference. There are various types of R&S procedures, such as indifference-zone [11], uncertainty evaluation [12], and optimal computing budget allocation (OCBA) [13]. Among them, OCBA is used in many fields owing to its excellent efficiency, simplicity, and strong theoretical background. It allocates the simulation budget to asymptotically maximize the lower bound of the probability of correct selection based on the ratio of the sample mean to sample variance. Based on the merits of OCBA, Jia et al. [14] applied it to improve the efficiency of SBPI and efficiently find the best action for a given state. Wu et al. [15] developed a sample path sharing procedure to further improve upon the above work. For a given state, a sample path is obtained by selecting an action and thereafter following the given policy. When the number of sample paths increases, the overlaps between the sample paths generated by different actions allow accurate estimation of the Q-value for each action. They reported that the sample path sharing procedure dramatically improves the efficiency of SBPI compared to the previous methods [14].

To derive the optimal policy, the SBPI should traverse all states until the base policy of each state can no longer be improved. However, the above works focus on improving the efficiency of the SBPI in a single state; i.e., there is room to further improve the efficiency of finding the optimal policy with SBPI. In this work, we propose a method to improve the overall efficiency of SBPI using OCBA and sample accumulation. Specifically, the proposed method accumulates simulation samples across all states to estimate the unknown state transition probabilities. These probabilities are used to estimate the mean and variance of the Q-value for each action. As the SBPI traverses the states, the probabilities become more accurate, thereby enabling precise estimations of the mean and variance. They allow the OCBA to allocate a budget suited to each action and to select the best action for a low cost. Thus, the proposed method can reduce the total budget required to derive the optimal policy with SBPI compared to other methods, which is demonstrated using two

MDP examples. The present work was adapted from our previous work [16], which used accumulated samples to estimate the mean of the Q-values. We expanded the use of these samples to further estimate the variance of the Q-values by fully adapting to the OCBA workflow.

## 2. Problem Definition

Herein, we consider a discrete-time MDP with discrete state space $S$ and discrete action space $A$. In the MDP, the policy learner or decision-maker is called the agent. Assume that the agent is at state $s$ and performs an action $a$. Then it transits to a new state in the set of reachable states $S_s^a = \{s_1, s_2, s_3\}$ that are determined by the unknown transition probabilities $P_{ss_1}^a, P_{ss_2}^a, P_{ss_3}^a$. The agent receives one of the numerical rewards $r(s_1), r(s_2), r(s_3)$ based on the state of arrival. We define a random variable $h(s, a)$, whose possible outcomes are the rewards received when arriving at the corresponding state by taking the action $a$:

$$h(s, a) \in \{r(s_1), r(s_2), r(s_3)\}. \tag{1}$$

If the transition probabilities are known, the expected reward for the action $a$ in state $s$ can be calculated as:

$$E[h(s, a)] = P_{ss_1}^a \cdot r(s_1) + P_{ss_2}^a \cdot r(s_2) + P_{ss_3}^a \cdot r(s_3). \tag{2}$$

Now, we formulate the policy improvement problem. In this study, we only consider a deterministic stationary policy $\pi$ (i.e., a mapping from $S$ to $A$), which is a guideline for the agent for the action that should be taken in a particular state. Assume that there exists a base policy $\pi$. For a given state $s$, if an action $a$ is taken and then the base policy $\pi$ is followed afterward, the Q-value of the action $a \in A$ can be defined as:

$$Q_\pi(s, a) = \lim_{T \to \infty} \left\{ E[h(s, a)] + E_\pi \left[ \sum_{t=1}^{T-1} \gamma^t h(s^t, \pi(s^t)) \middle| s, a \right] \right\}, \tag{3}$$

where $T$ is the terminal time index, $\gamma \in [0, 1]$ is the discount rate, and $s^t$ is one of the reachable states at time $t$ (i.e., $s^t \in S_{s^{t-1}}^{\pi(s^{t-1})}$). $E_\pi$ indicates the expected sum of discounted rewards obtained by taking actions following the given policy $\pi$ from time 1 to $T - 1$. Using the definition above, the policy improvement at state $s$ can be defined as:

$$\pi_{PI}(s) = a_b = \arg\max_{a \in \{a_1, a_2, \dots, a_k\}} Q_\pi(s, a), \tag{4}$$

where $\pi_{PI}$ is the improved policy from $\pi$ by updating the previous action $\pi(s)$ with the best action $a_b$ in $s$. In practice, the transition probabilities are usually unknown, and $Q_\pi(s, a)$ cannot be calculated directly. Thus, Equation (3) can only be calculated using an infinite number of simulation replications $n$; i.e.,

$$Q_\pi(s, a) = \lim_{T \to \infty} \lim_{n \to \infty} \frac{1}{n} \sum_{n \in \mathbb{N}} \left\{ h(s, a) + \left[ \sum_{t=1}^{T-1} \gamma^t h(s^t, \pi(s^t)) \middle| s, a \right] \right\}. \tag{5}$$

Since it is practically infeasible to take actions infinitely in a simulation replication, $T$ becomes a decision variable called epoch, and Equation (5) is approximated by

$$Q_\pi^T(s, a) = \lim_{n \to \infty} \frac{1}{n} \sum_{n \in \mathbb{N}} \left\{ h(s, a) + \left[ \sum_{t=1}^{T-1} \gamma^t h(s^t, \pi(s^t)) \middle| s, a \right] \right\}. \tag{6}$$

That is, in a single replication, $T$ actions, including action $a$ from state $s$, are sequentially taken depending on $\pi$, and a sample trajectory of $Q_\pi^T(s, a)$ can be obtained as follows:

$$\dot{Q}_\pi^T(s,a) = h(s,a) + \left[ \sum_{t=1}^{T-1} \gamma^t h(s^t, \pi(s^t)) \middle| s, a \right]. \tag{7}$$

In practice, the number of simulation replications $n$ is typically limited; thus, $Q_\pi^T(s, a)$ can be estimated from the average of the sample trajectories:

$$\bar{Q}_\pi^T(s,a) = \frac{1}{n} \sum_{l=1}^{n} \dot{Q}_\pi^{T,l}(s,a), \tag{8}$$

when $n$ is large, due to the central limit theorem, it is reasonable to assume that $\bar{Q}_\pi^T(s, a)$ follows a normal distribution of $Q_\pi^T(s, a)$ [17].

For each $s \in S$, the SBPI estimates $\bar{Q}_\pi^T(s, a)$ for every available action using many simulation replications and improves $\pi$ by replacing the base action $\pi(s)$ with the estimated best action $a_e$:

$$\pi_{SBPI}(s) = a_e = \arg\max_{a \in \{a_1, a_2, \ldots, a_k\}} \bar{Q}_\pi^T(s, a). \tag{9}$$

To improve $\pi$ exactly using the SBPI, the selection of $a_e$ should be correct (i.e., $a_e = a_b$) at each $s$. From this point of view, the probability of the correct selection $P\{CS\}$ can be defined according to [14] as

$$\begin{aligned} P\{CS\} &= P\{a_e = a_b\} \\ &= P\{\tilde{Q}_\pi(s, a_e) \geq \tilde{Q}_\pi(s, a) - \epsilon\}. \end{aligned} \tag{10}$$

Here, $\epsilon \geq 0$ is the tolerance level, and $\tilde{Q}_\pi(s, a)$ is the posterior distribution of $Q_\pi(s, a)$. Increasing $n$ and $T$ for each action can maximize $P\{CS\}$, but it causes the efficiency problem of the SBPI, as mentioned earlier.

To solve the problem, the existing methods [14,15] apply OCBA to allocate a given simulation budget $N$ efficiently; i.e.,

$$\arg\max_{\{n_1, \ldots n_k, T\}} P\{CS\},$$
$$\text{s.t.} \sum_{i=1}^{k} n_i T = N, \text{ and } n_i \geq 0, \tag{11}$$

where $n_i$ is the number of simulation replications allocated to estimate the Q-value of the $i$th action. Under this definition, the OCBA aims to accurately allocate $N$ to each action so that the best action can be correctly selected with higher $P\{CS\}$, thereby improving the efficiency of SBPI. The allocation rule of OCBA [14] is defined as follows:

$$\frac{n_i}{n_j} = \left[ \frac{\sigma_i / (\delta_{e,i}^T + \epsilon - 2c)}{\sigma_j / (\delta_{e,j}^T + \epsilon - 2c)} \right]^2,$$
$$n_e = \sigma_e \sqrt{\sum_{i=1, i \neq e}^{k} \left( \frac{n_i}{\sigma_i} \right)^2}, \quad i \neq j \neq e, \tag{12}$$

where $\delta_{e,i}^T \equiv \bar{Q}_\pi^T(s, a_e) - \bar{Q}_\pi^T(s, a_i)$, $\sigma_i^2$ is the variance of Q-value for action $a_i$, and $0 \leq c \leq \epsilon/2$ is a constant determined by $\epsilon$. In practice, $\sigma_i^2$ is unknown in advance and so is approximated by sample variance [13]. In Equation (11), $T$ is a decision variable, which is an important hyperparameter to determine the optimal simulation length while ensuring that the estimation of the action is as close as possible to the estimation with infi-

nite simulation length. To determine the optimal $T$ for each simulation sample, the authors of [14] proposed

$$T = \left\lceil \frac{\log\left[\frac{c(1-\gamma)}{F}\right]}{\log \gamma} \right\rceil, \tag{13}$$

where $\lceil \cdot \rceil$ is the ceiling function and $F$ is the maximum absolute reward; i.e., $\max_{s \in S}|r(s)|$.

To find the optimal policy through SBPI, it is necessary to repeat the SBPI and traverse across states until the policy is no longer improved. Here, the quality of the policy can be evaluated as

$$\mathcal{V}_\pi(s_\alpha) = \lim_{T \to \infty} E_\pi \left[ \sum_{t=0}^{T-1} \gamma^t h\left(s^t, \pi(s^t)\right) \Big| s_\alpha \right], \tag{14}$$

where $\mathcal{V}_\pi(s_\alpha)$ is the expected sum of discounted reward obtained by sequentially taking actions based on $\pi$ from the initial state $s_\alpha$. $\mathcal{V}_\pi(s_\alpha)$ is maximized if $\pi$ is the optimal policy $\pi^*$. The iteration number of SBPI is denoted as $m$. When the total simulation budget is given as $B$, the problem of finding $\pi^*$ using SBPI can be defined as

$$\max \mathcal{V}_{\pi_{PI}^m}(s_\alpha) \quad \text{s.t. } Nm = B, \tag{15}$$

where $\pi_{PI}^m$ represents the improved policy via $m$th SBPI. If $N$ increases, the existing methods [14,15] can correctly select the best action in each state and improve the policy. However, before $\pi$ converges to $\pi^*$, the selected best action may not be the actual best action. In other words, the best action in the same state may change as the policy is updated, as shown in Equation (9). When $B$ is fixed, increasing $N$ causes insufficient iterations of the SBPI. Hence, regardless of how correctly the best action is selected in each state, the existing methods may not converge to $\pi^*$. Considering the this issue, it is necessary to decrease $N$ and increase $m$ to find the optimal policy. However, the existing methods may not be able to correctly select the best action when $N$ is small, since they discard the previous measurements after each update. In the next section, we propose a method for accumulating simulated samples to allow the OCBA to allocate small $N$ efficiently and select the best action correctly.

### 3. Proposed Method

Herein, we illustrate the proposed method in two parts. Firstly, we show how to utilize accumulated samples to estimate the unknown transition probabilities. Secondly, we utilize the probability estimates to derive the mean and variance of the Q-values for OCBA to accurately allocate the budget. As defined in Equation (3), if the transition probabilities are known, they can be unrolled as

$$
\begin{aligned}
Q_\pi(s,a) = {} & \sum_{s' \in S_s^a} P_{ss'}^a r\left(s'\right) + \gamma \sum_{s' \in S_s^a} P_{ss'}^a \sum_{s'' \in S_{s'}^{\pi(s')}} P_{s's''}^{\pi(s')} r\left(s''\right) + \ldots \\
& + \gamma^{T-1} \sum_{s' \in S_s^a} P_{ss'}^a \cdots \sum_{s'^{(T)} \in S_{s'^{(T-1)}}^{\pi(s'^{(T-1)})}} P_{s'^{(T-1)}s'^{(T)}}^{\pi(s'^{(T-1)})} r\left(s'^{(T)}\right).
\end{aligned} \tag{16}
$$

The formulation above can be further rewritten in a recursive form by extracting the common factor $\sum_{s' \in S_s^a} P_{ss'}^a$:

$$Q_\pi(s,a) = \sum_{s' \in S_s^a} P_{ss'}^a \left[ r(s') + \gamma Q_\pi(s', \pi(s')) \right]. \tag{17}$$

With Equation (17), $Q_\pi(s,a)$ can be computed for all the action candidates in state $s$, and the best action can be accurately selected using Equation (4). However, the transition prob-

abilities are unknown beforehand. To this end, our method accumulates all the state–action pairs generated by each simulation sample to estimate the unknown transition probabilities.

Let $N_{ss'}^a$ be the cumulative number for arriving at a state $s'$ when an action $a$ is taken in that state $s$. Then, the transition probability can be estimated as

$$\hat{P}_{ss'}^a = \frac{N_{ss'}^a}{\sum_{s_i \in S_s^a} N_{ss_i}^a}. \tag{18}$$

We use a table to store all the $N_{ss'}^a$ for estimating and updating probabilities, as shown in Figure 2. The first column in Figure 2 represents the possible state–action pairs. The first row represents the reachable states.

|  From state & action taken | $s_1$ | $s_2$ | ... | $s_n$ |
|---|---|---|---|---|
|  | | **To state** | | |
| $(s_1, a_1)$: | $N_{s_1 s_1}^{a_1}$ | $N_{s_1 s_2}^{a_1}$ | ... | $N_{s_1 s_n}^{a_1}$ |
| $(s_1, a_2)$: | 0 | $N_{s_1 s_2}^{a_2}$ | ... | 0 |
| $\vdots$ | $\vdots$ | $\vdots$ | $\vdots$ | $\vdots$ |
| $(s_n, a_n)$: | $N_{s_n s_1}^{a_n}$ | 0 | ... | $N_{s_n s_n}^{a_n}$ |

**Figure 2.** Table of the cumulative number of state–action pairs, where 0 represents the unreachable states.

As the simulation progresses, the state–action pairs accumulate and result in accurate estimates of the transition probability. Thus, we use these estimates to derive the mean and variance of the Q-values, which allows OCBA accurately allocate the given simulation budget to each available action based on the sample accumulation. To derive the mean with estimated probabilities, we substitute Equation (18) into (17):

$$\hat{Q}_\pi(s,a) = \sum_{s' \in S_s^a} \hat{P}_{ss'}^a \left[ r(s') + \gamma \hat{Q}_\pi(s', \pi(s')) \right]. \tag{19}$$

When the samples are accumulated, the probability estimates are approximate equal to the real probability distribution, thereby ensuring that $\hat{Q}_\pi(s,a)$ is an unbiased estimated mean of the Q-value. For the variance of the Q-value, we utilize the variance definition of a random variable as follows:

$$\sigma_\pi^2(s,a) = E_\pi \left[ \left( h(s,a) + \sum_{t=1}^{T-1} \gamma^t h(s^t, \pi(s^t)) \right)^2 \Bigg| s,a \right] - Q_\pi(s,a)^2. \tag{20}$$

Unrolling the quadratic sum in Equation (20), we then have

$$\sigma_\pi^2(s,a) = \sum_{s' \in S_s^a} P_{ss'}^a r(s')^2 + 2 \sum_{s' \in S_s^a} P_{ss'}^a r(s') E_\pi \left[ \sum_{t=1}^{T-1} \gamma^t h(s^t, \pi(s^t)) \Bigg| s', \pi(s') \right]$$
$$+ \sum_{s' \in S_s^a} P_{ss'}^a E_\pi \left[ \left( \sum_{t=1}^{T-1} \gamma^t h(s^t, \pi(s^t)) \right)^2 \Bigg| s', \pi(s') \right] - Q_\pi(s,a)^2. \tag{21}$$

From the equation above, we can observe that the first term $E_\pi \left[ \sum_{t=1}^{T-1} \gamma^t h(s^t, \pi(s^t)) \right]$ has a form similar to Equation (3). The difference here is that it multiplies $\gamma$ from the first state and follows the base policy from the beginning. Further, the second term

$E_\pi\left[\left(\sum_{t=1}^{T-1}\gamma^t h(s^t,\pi(s^t))\right)^2\right]$ has a form similar to the expected form in Equation (20). Thus, we extract $\gamma$ from the first term and $\gamma^2$ from the second term

$$\sigma_\pi^2(s,a) = \sum_{s'\in S_s^a} P_{ss'}^a r(s')^2 + 2\gamma \sum_{s'\in S_s^a} P_{ss'}^a r(s') E_\pi\left[\sum_{t=1}^{T-1}\gamma^{t-1}h(s^t,\pi(s^t))\,\Big|\,s',\pi(s')\right]$$
$$+\gamma^2 \sum_{s'\in S_s^a} P_{ss'}^a E_\pi\left[\left(\sum_{t=1}^{T-1}\gamma^{t-1}h(s^t,\pi(s^t))\right)^2\,\Big|\,s',\pi(s')\right] - Q_\pi(s,a)^2. \tag{22}$$

It is clear that the term $E_\pi\left[\sum_{t=1}^{T-1}\gamma^{t-1}h(s^t,\pi(s^t))\right]$ can be substituted by $Q_\pi(s',\pi(s'))$ based on Equation (3), and the term $E_\pi\left[\left(\sum_{t=1}^{T-1}\gamma^{t-1}h(s^t,\pi(s^t))\right)^2\right]$ can be rewritten as $\sigma_\pi^2(s',\pi(s')) + Q_\pi(s',\pi(s'))^2$ according to Equation (20). Thus, we have

$$\sigma_\pi^2(s,a) = \sum_{s'\in S_s^a} P_{ss'}^a [\gamma^2 Q_\pi(s',\pi(s'))^2 + 2\gamma r(s')Q_\pi(s',\pi(s')) + r(s')^2]$$
$$+\gamma^2 \sum_{s'\in S_s^a} P_{ss'}^a \sigma_\pi^2(s',\pi(s')) - Q_\pi(s,a)^2 \tag{23}$$
$$= \sum_{s'\in S_s^a} P_{ss'}^a \left[r(s') + \gamma Q_\pi(s',\pi(s'))\right]^2 + \gamma^2 \sum_{s'\in S_s^a} P_{ss'}^a \sigma_\pi^2(s',\pi(s')) - Q_\pi(s,a)^2.$$

To simplify the equation above, we rewrite it in recursive form. Let $R(s')$ be a new reward function

$$R(s') = \left[r(s') + \gamma Q_\pi(s',\pi(s'))\right]^2 - \frac{Q_\pi(s,a)^2}{\sum_{s'\in S_s^a} P_{ss'}^a}. \tag{24}$$

Then, Equation (23) can be rewritten with Equation (24) as:

$$\sigma_\pi^2(s,a) = \sum_{s'\in S_s^a} P_{ss'}^a \left[R(s') + \gamma^2 \sigma_\pi^2(s',\pi(s'))\right]. \tag{25}$$

As the unknown transition probabilities can be calculated from the table, as shown in Figure 2, the variance of the Q-values can be estimated as

$$\hat{\sigma}_\pi^2(s,a) = \sum_{s'\in S_s^a} \hat{P}_{ss'}^a \left[\hat{R}(s') + \gamma^2 \hat{\sigma}_\pi^2(s',\pi(s'))\right], \tag{26}$$

Despite the fact that the $\hat{\sigma}_\pi^2(s,a)$ of each action's Q-value may not be accurate at the beginning of the iteration, the allocation rule of the OCBA enables additional simulation replications to the promising action as the number of iterations increases by updating the measurement for each action. Thus, the estimation of probabilities becomes accurate as more samples are accumulated and results in an accurate estimate of the variance. To estimate $\hat{Q}_\pi(s,a)$ and $\hat{\sigma}_\pi^2(s,a)$, we rewrite them into recursive forms. Since it is not feasible to use an infinite number of $T$ for estimation, we use Equation (13) to determine $T$. Then, $\hat{Q}_\pi(s,a))$ and $\hat{\sigma}_\pi^2(s,a)$ are approximated as

$$\hat{Q}_\pi^T(s,a) = \sum_{s'\in S_s^a} \hat{P}_{ss'}^a \left[r(s') + \gamma \hat{Q}_\pi^{T-1}(s',\pi(s'))\right], \tag{27}$$

$$\hat{\sigma}_\pi^{2,T}(s,a) = \sum_{s'\in S_s^a} \hat{P}_{ss'}^a \left[R^T(s') + \gamma^2 \hat{\sigma}_\pi^{2,T-1}(s',\pi(s'))\right]. \tag{28}$$

In the existing methods, the simulation budget allocated to estimate each $\bar{Q}_\pi^T(s,a)$ in a given state is limited to $N$. Thus, when $N$ is small, inaccurate estimates of $\bar{Q}_\pi^T(s,a)$ and sample variance degrade the effectiveness of OCBA at allocating simulation replications to each action and result in low efficiency of SBPI. On the other hand, the proposed method accumulates simulation samples from the previous $m$ updates (i.e., $\sum_1^m N$) to estimate and update transition probabilities. These probabilities are then used to compute $\hat{Q}_\pi^T(s,a)$ and $\hat{\sigma}_\pi^{2,T}(s,a)$ for the OCBA allocation rule of Equation (12), so that the OCBA can accurately allocate $N$ to each action in each state and help improve the overall efficiency of SBPI. The proposed method is summarized in Algorithm 1.

---

**Algorithm 1** Efficient simulation-based policy improvement with optimal computing budget allocation based on accumulated samples.

---

**Require:** a base policy $\pi$, an incremental replication $\triangle$, simulation budget $N$, an initial state $s_a \in S$, total simulation budget $B$. Initialize the probability table. Set the iteration number of SBPI $m \rightarrow 1$. Determine $T$ using Equation (13).

1: **while** $Nm \leq B$ **do**
2: 　　**for** $s$ in $S$ **do**
3: 　　　　Initialize $l \rightarrow 0$
4: 　　　　**if** SBPI has never been applied to $s$ **then**
5: 　　　　　　Set $n_1^l = \cdots = n_k^l = n_0$
6: 　　　　　　Run $n_0$ for $a \in \{a_1, \cdots, a_k\}$, and store $N_{ss'}^a$
7: 　　　　　　Estimate $\hat{P}_{ss'}^a$ by (18) and Calculate $\hat{Q}_\pi^T(s,a)$, $\hat{\sigma}_\pi^{2,T}(s,a)$ for each $a$
8: 　　　　**else**
9: 　　　　　　Calculate $\hat{Q}_\pi^T(s,a)$, $\hat{\sigma}_\pi^{2,T}(s,a)$ by $\hat{P}_{ss'}^a$ and
10: 　　　　　　set $n_1^l = \cdots = n_k^l = 0$
11: 　　　　**end if**
12: 　　　　Select $a_e \leftarrow \arg\max_{a \in \{a_1, a_2, \ldots a_k\}} \hat{Q}_\pi^T(s,a)$
13: 　　　　**while** $\sum_{i=1}^k n_i^l T < N$ **do**
14: 　　　　　　Increase the simulation budget by $\triangle$
15: 　　　　　　Compute new allocation $n_1^{l+1}, \ldots, n_k^{l+1}$ with $\hat{Q}_\pi^T(s,a)$ and $\hat{\sigma}_\pi^{2,T}(s,a)$ using (12)
16: 　　　　　　Run additional $\max(0, n_i^{l+1} - n_i^l)$ simulations for each $a$
17: 　　　　　　Update $\hat{P}_{ss'}^a$ and compute $\hat{Q}_\pi^T(s,a)$, $\hat{\sigma}_\pi^{2,T}(s,a)$
18: 　　　　　　Select $a_e \leftarrow \arg\max_{a \in \{a_1, a_2, \ldots a_k\}} \hat{Q}_\pi^T(s,a)$ and set $l \leftarrow l+1$
19: 　　　　**end while**
20: 　　　　**return** $\pi_{PI}(s) \leftarrow a_e$
21: 　　　　**if** $Nm \geq B$ **then**
22: 　　　　　　**break**
23: 　　　　**else**
24: 　　　　　　$m \leftarrow m+1$
25: 　　　　**end if**
26: 　　**end for**
27: **end while**

---

In Algorithm 1, lines 12 to 16 show the procedure of OCBA using $\hat{Q}_\pi^T(s,a)$ and $\hat{\sigma}_\pi^{2,T}(s,a)$ to allocate the simulation budget for each action. It is noted that Algorithm 1 initially allocates $n_0$ simulation replications to each action only when the SBPI has never been applied to that state, whereas the existing methods allocate $n_0$ to each action regardless of whether the state has been visited. The reason for this is that our method stores the accumulated samples from the previous updates and calculates the prior information for the state (i.e., line 5, 6); it does not allocate $n_0$ for the prior information in the next visit. For SBPI, the estimated best action $a_e$ is considered as the best action for a given state and is used to update the base policy. If the base policy is updated as the SBPI proceeds, the previously selected $a_e$ may no longer be the best for the updated policy even in the same state. Owing

to this property of SBPI, existing methods may waste some of the simulation budget for previous policy updates when the SBPI is applied to each state to obtain the optimal policy. To avoid this, Algorithm 1 accumulates simulation samples from the previous updates and applies them to compute $\hat{Q}_\pi^T(s, a)$ and $\hat{\sigma}_\pi^{2,T}(s, a)$,a) to accurately deduce the optimal action in the following $m$. As Algorithm 1 proceeds, the accumulated samples help SBPI obtain the optimal policy with minimum $m$, thereby reducing the total simulation budget required. This suggests that the proposed method is efficient for improving the overall efficiency of SBPI, which accumulates the samples considering the property of SBPI.

The proposed method can be computationally inefficient owing to the recursive forms of $\hat{Q}_\pi^T(s, a)$ and $\hat{\sigma}_\pi^{2,T}(s, a)$. However, this issue is tolerable compared to the simulation cost of an actual system. For a real-world MDP system (e.g., water resource management), a large duration of time is needed to return a reward. In this aspect, the computational time taken by $\hat{Q}_\pi^T(s, a)$ and $\hat{\sigma}_\pi^{2,T}(s, a)$ is negligible relative to the running time of obtaining the reward. Moreover, compared with the existing methods, the proposed method ha higher efficiency for finding the optimal policy, as shown in the Experiments section. This indicates that the superior efficiency of the proposed method is sufficient to mitigate its limitation with respect to the recursive calculation.

## 4. Experiments

Herein, we compare our method with five existing methods (EA, OCBAPI [14], OCBA-S [15], EA-sample accumulation (SA), and OCBAPI-SA) using two MDP models, namely, a two-state example and its extended version. The extended version is used to verify the effectiveness and efficiency of the proposed method in a more complex manner. The description of the five methods is summarized in Table 1.

**Table 1.** Summary of comparison methods.

| Method | Description | Allocation Rule | Best Action Selection |
|---|---|---|---|
| EA | Standard SBPI | $n_i = N/k$ | $a_e = \arg\max_a \bar{Q}_\pi^T(s, a)$ |
| OCBAPI | Using OCBA to allocate the simulation budget efficiently | Equation (12) | $a_e = \arg\max_a \bar{Q}_\pi^T(s, a)$ |
| OCBA-S | Improving efficiency of OCBAPI with sample path sharing | Equation (12) | $a_e = \arg\max_a \bar{Q}_\pi^T(s, a)$ [a] |
| EA-SA | Using sample accumulation for EA to select the best action via Equation (27) | $n_i = N/k$ | $a_e = \arg\max_a \hat{Q}_\pi^T(s, a)$ |
| OCBAPI-SA | Using sample accumulation for OCBAPI to select the best action via Equation (27) | Equation (12) | $a_e = \arg\max_a \hat{Q}_\pi^T(s, a)$ |
| OCBAPI-SA2 (Algorithm 1) | Using the estimated mean from Equation (27) and variance from Equation (28) of the Q-value to efficiently allocate computing budget for OCBAPI-SA. | use $\hat{Q}_\pi^T(s, a)$, $\hat{\sigma}_\pi^{2,T}(s, a)$ for Equation (12) | $a_e = \arg\max_a \hat{Q}_\pi^T(s, a)$ |

[a] Sample path sharing is used to calculate $\bar{Q}_\pi^{T,S}(s, a)$; see more details in [15].

The two-state example was expanded by increasing the available actions in state $s_2$, as shown in Figure 3. The example had two states, and 20 actions were available in each state. Action $\alpha_i$ and $\beta_i$ were the $i$th elements in the action vector $A = B = [0.0, 0.05, \ldots, 0.95]$. The state transition probabilities were determined by the choice of actions. For example, if the agent selected an action $\alpha_2$ in state $s_1$, the probability of remaining in state $s_1$ was 0.05 and the probability of transferring to state $s_2$ was 0.95. When the agent arrived at state $s_1$, it always received a reward of 0 but received a reward of 1

when arriving at state $s_2$. The extended version of the two-state example was obtained by increasing the number of states from 2 to 10, as shown in Figure 4. The agent in the extended model received a reward of 5 only when it reached state $s_{10}$.

The base policy was set as selecting the action 0.5 in each state for two examples, and state $s_1$ was set as the initial state. In the two-state example, let the discount rate $\gamma = 0.7$ and tolerance level $\epsilon = 0.1$. Then, $c = \epsilon/2 = 0.05$ is obtained and $T = \lceil (\log [c(1-\gamma)/F]) / \log \gamma \rceil = 12$, where $F = 1$. The simulation budget for each method in each state was $N = 60T$. The number of iterations of the SBPI was $m = 20$ so that the total simulation budget was $B = Nm = 1200T$. In the extended example, we only changed the tolerance level to $\epsilon = 0.5$ and the iteration number to $m = 100$. For the methods using OCBA, we set $n_0 = 2T$, and incremental replication $\triangle = 2T$. The value function and PCS of each method were estimated over 5000 independent replicated experiments, and the results are shown in Figure 5. In both examples, the proposed OCBAPI-SA2 converged to the optimal policy faster than the other methods, and the gap increased as the problem complexity increased. All experiments were implemented based on Python (version 3.7.9).

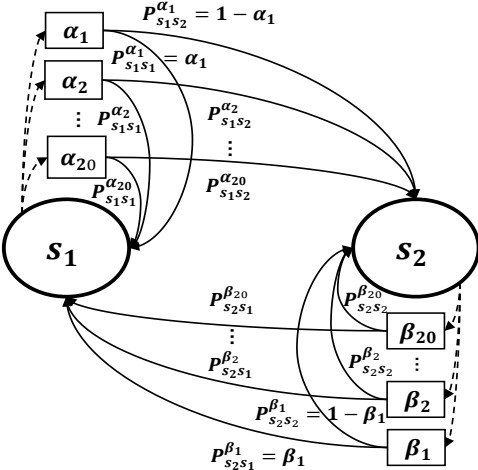

**Figure 3.** A two-state Markov decision process.

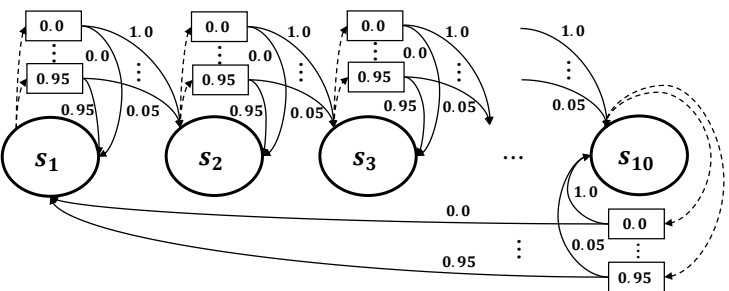

**Figure 4.** An extended version of the two-state example.

The results of EA-SA, OCBAPI-SA, and OCBAPI-SA2 indicate the effectiveness of the sample accumulation in the SBPI. As shown in Figure 5A,C, these methods had superior efficiencies to their original versions, i.e., EA and OCBA. As $m$ increased, the accumulated samples allowed precise estimates of $\hat{Q}_\pi^T(s, a)$, which enabled the methods to select the best action correctly, as shown in Figure 5B,D. Meanwhile, OCBA-S achieved a higher $P\{CS\}$ than EA-SA at the beginning iteration of SBPI, which can be attributed to its efficient allocation rule and sample path sharing. However, the sample path sharing was limited to a single state, EA-SA, and the sample accumulation improved as $m$ increased.

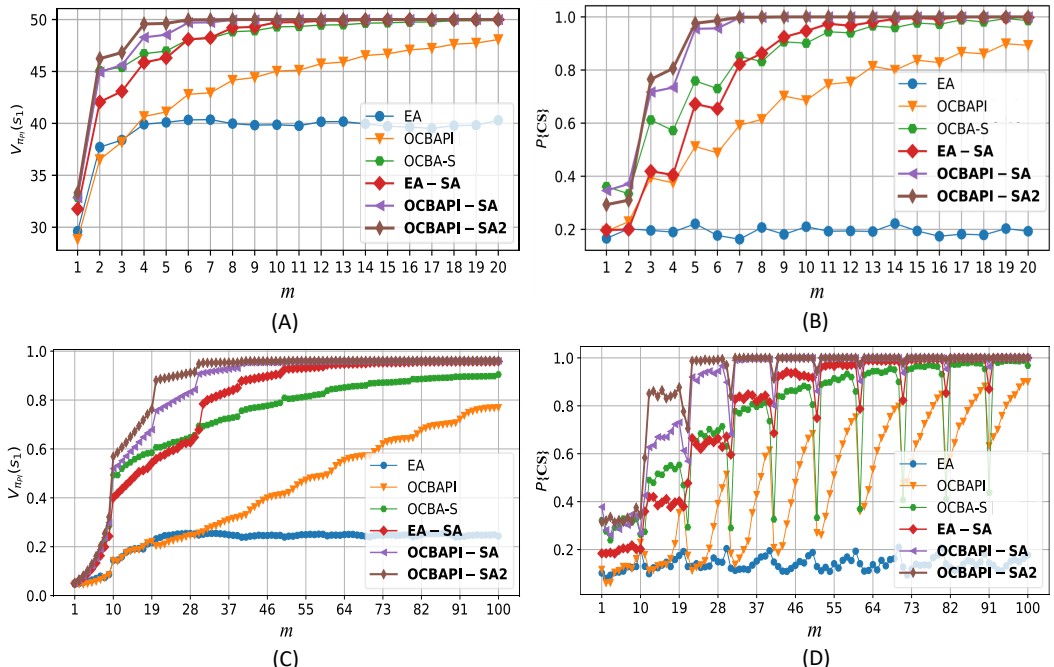

**Figure 5.** Graphs indicate the value function of the improved policy and $P\{CS\}$ of each method for the two examples: (**A**,**B**) the results of the two-state example; (**C**,**D**) the results of the extended version.

While OCBAPI-SA only used $\hat{Q}_\pi^T(s,a)$ and sample variance for OCBA allocation, OCBAPI-SA2 used $\hat{Q}_\pi^T(s,a)$ and $\hat{\sigma}_\pi^{2,T}(s,a)$ evaluated from the accumulated samples. When $N$ was small, OCBAPI-SA could not efficiently allocate $\triangle$ to each action owing to inaccurate sample estimates. On the contrary, OCBAPI-SA2 surpassed the limited $N$ by accumulating samples from previous updates to estimate $\hat{Q}_\pi^T(s,a)$ and $\hat{\sigma}_\pi^{2,T}(s,a)$. As SBPI proceeded, OCBAPI-SA2 could efficiently allocate $\triangle$ to the promising actions and resulted in a higher $P\{CS\}$ than OCBAPI-SA. Although the gap between them was relatively insignificant in the two-state example owing to the small number of states, it became large in the complex problem, as shown in Figure 5C,D.

## 5. Conclusions

In this study, we proposed a method called OCBAPI-SA2 that uses OCBA to improve the overall efficiency of SBPI. Unlike existing methods, OCBAPI-SA2 aims to improve the overall efficiency by considering the state traversal property of SBPI. To achieve this, OCBAPI-SA2 applies SBPI to traverse across states and accumulates the simulation samples to estimate the unknown transition probabilities. Then, it utilizes these probabilities to compute the mean and variance of the Q-values for OCBA to efficiently allocate simulation budget. With the accumulation of samples, OCBAPI-SA2 allows SBPI to obtain the optimal policy with a lower simulation budget, which is important in practice for complex systems with limited budgets. The experimental results show that the superior efficiency of the OCBAPI-SA2 is comparable to those of existing methods. Considering the properties of the SBPI, the sequence of improving policy has a significant impact on reducing the simulation budget. To further improve the overall efficiency of SBPI, our future work will focus on the optimal sequence of state traversal.

**Author Contributions:** X.H. and S.H.C. conceived the methodology and designed the experiments; X.H. conducted the experiments; X.H. and S.H.C. analyzed the experimental results; X.H. wrote the manuscript; X.H. and S.H.C. reviewed and revised the manuscript. All authors have read and agreed to the published version of the manuscript.

**Funding:** This work was supported by the Ewha Womans University Research Grant of 2022.

**Data Availability Statement:** The data used to support the findings of this study are available from the corresponding author upon request.

**Conflicts of Interest:** The authors declare that there are no conflict of interest regarding the publication of this paper.

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
