# Peer review of "An Efficient Simulation-Based Policy Improvement with Optimal Computing Budget Allocation Based on Accumulated Samples"

_electronics, doi:10.3390/electronics11071141_

Round 1
Reviewer 1 Report
The article presents an improvement in efficiency of simulation-based policy improvement through better allocation of computing budget by reusing accumulated samples. The usual approach to finding the optimal policy in a Markov decision process is to focus on a state s and action a, simulate n outcomes T timesteps in advance, and estimate Q-value from the rewards obtained. Optimal computing budget allocation (OCBA) serves to efficiently allocate simulation replications to various actions for a given state s – there is no point in more accurately quantifying Q of an action which is already accurately known to be poor, it is better to focus on the ones that look good but are poorly known and may thus be optimal with a high probability. The presented improvement in efficiency is, IF I UNDERSTAND CORRECTLY, in not using the simulation replication just for the state s from which it starts but to use every sample that happens to be taken at a given state s' when deducing the transition probabilities from the state s'. For example, if T = 10, each simulation replication from the state s consists of 10 transitions starting from 10 (possibly different) states. Why not use them all when deducing the probabilities?
The idea is good, but somewhat obvious. Why would one not use all the transitions?
To me, the manuscript is realtively complicated, unclear, hard to understand. I'd expect it to require some effort as I'm not an expert in the exact subfield; however, my impression is that it could, and should, be made easier to follow.
My main concern is the applicability of the method as a whole in any real-world example. By definition of the problem, transition probabilities are unknown, yet you simulate the process. How can you? With what probability do you pick the transition ss1 in the Markov process simulation when the probability of the transition is unknown? Contrary to the assumptions of the method, the transition probabilities in the provided numerical examples are specified (so they could be solved analytically without any simulation).
The paragraph in lines 84–94 seems to be confusing two different issues. One is the computational efficiency that is the topic of the manuscript. The other one is convexity / local minima. I do not think the problem of seeking the optimal policy as presented is convex. The action a at a given state s may be optimal at the given policy for the other states, and it may be so for every s in S, but the overall policy may still be suboptimal. It may be possible to find a better policy than the given one even if no change in policy for a single s in S improves the policy. In spite of what you say, your algorithm does not cure this problem. The algorithm circles through s in S and if no change in policy for a single s will improve overall policy, no improvement will be found.
Both presented numerical examples are extremely simple, convex, the best policy at any given state is independent of the policies at other states. That is an unfortunate coincidence.
For equation (26), you say that it is clear that it is an unbiased estimator for variance. How can you be so confident without demonstrating it? In the next sentence, you claim it is not feasible to use an infinite T. Why not? The number of states is finite, so the same state will be re-visited in a finite number of steps and recursion can be utilised.
Some of the dilemmas would be clarified if the source code was available. The Instructions for Authors https://www.mdpi.com/journal/electronics/instructions#suppmaterials state that "For work where novel computer code was developed, authors should release the code". Please provide the code as required.
Matija Perne
Author Response
Dear Reviewer 1
Thank you for considering this manuscript and suggesting constructive comments.
Your review has greatly helped to improve the quality of this manuscript.
We hope that our revised manuscript and the responses to your comments will meet your expectations.
You can see the responses in the attached file and check these revisions in the attached highlighted document.

Reviewer 2 Report
Please consider attached file.

Author Response

(The authors gave the same response as above.)

Round 2
Reviewer 1 Report
The presented algorithm serves to develop a decision-making policy when transition probabilities from a state to another state given a control action are not known. The transition probabilities are estimated through sampling in simulation, and the best action in each state is determined.
The new version of the manuscript and the authors' response to my previous comments confirm my belief that the method as presented cannot work and does not make sense. I therefore recommend the paper not to be published until the method is either corrected or shown to work.
The step of estimating the transition probability through sampling is not reasonable, it is cyclic. One cannot sample a random variable of an unknown probability distribution. If the probability distribution is not known, the method cannot be used, and if it is known, the method does not need to be used, as there is no need to estimate what is already known.
The two presented numerical examples illustrate the problem. While the manuscript emphasises numerous times that the transition probabilities are unknown, in the examples they ARE known, they are listed in lines 161–164. They have to be known for the method to work. Yet if they are known, the method is not necessary. The method is pointless.
Author Response
We would like to express sincere appreciation for your time and efforts on this manuscript. Also, we apologize for the inconvenience you experienced during the review process. Please check the attached our responses to your comments. We hope that this response will meet your expectations.

Reviewer 2 Report
The authors have satisfactorily responded to all my questions and made the necessary changes to the manuscript.
Author Response
We would like to express sincere appreciation for your time and efforts on this manuscript. Your review has greatly helped to improve the quality of this manuscript. Also, thanks for your positive decision.
This manuscript is a resubmission of an earlier submission. The following is a list of the peer review reports and author responses from that submission.